# Effect of Multiple Enamel Surface Treatments on Micro-Shear Bond Strength

**DOI:** 10.3390/polym13203589

**Published:** 2021-10-18

**Authors:** René Daher, Ivo Krejci, Mustapha Mekki, Charlotte Marin, Enrico Di Bella, Stefano Ardu

**Affiliations:** 1Division of Cardiology & Endodontology, University Clinics of Dental Medicine, University of Geneva, 1211 Geneva, Switzerland; rene.daher@unige.ch (R.D.); ivo.krejci@unige.ch (I.K.); 2Division of Fixed Prosthodontics and Biomaterials, University Clinics of Dental Medicine, University of Geneva, 1211 Geneva, Switzerland; mustapha.mekki@unige.ch; 3Independent Researcher, 1211 Geneva, Switzerland; marin.charlotte@ymail.com; 4Department of Economics and Quantitative Methods, University of Genoa, 16100 Genoa, Italy; edibella@unige.it

**Keywords:** etching time, bonding, adhesion

## Abstract

**Objectives**: The aim of this study was to evaluate the effect of different enamel surface treatments on the micro-shear bond strength. **Materials and methods**: Sixty-four approximal surfaces from freshly extracted molars were randomly assigned to eight groups, according to combinations of the following enamel surface treatments: ground or unground, 37.5% phosphoric acid etching time of 15 or 30 s, and with or without primer application. The same bonding agent (Optibond FL™ Adhesive, Kerr) was then used for all groups, and a 1.8 mm diameter resin composite (Harmonize™, Kerr) cylinder was built up on the bonded surface. Samples underwent a shear force test at a crosshead speed of 1 mm/min until failure. Bond strength was calculated, and failure modes were inspected under an optical microscope. **Results**: Bond-strength values ranged from 8.2 MPa for 15 s etched unground enamel with primer application to 19.6 MPa for 30 s etched ground enamel without primer application. ANOVA and Fisher’s LSD post hoc tests revealed significant differences between the groups. **Conclusions**: Etching time and grinding have a statistically significant effect on the micro-shear bond strength of a three-step etch-and-rinse adhesive system on enamel. Primer application does not seem to be beneficial for enamel adhesion.

## 1. Introduction

Long-term success of direct bonded restorations is based on several factors, such as control of the humidity of the operatory field, a correct composite layering technique, sufficient polymerization, and a scrupulous execution of the adhesive protocols. The majority of these factors are widely established in the literature [1,2,3,4,5,6,7,8], while some steps in adhesive protocols are still a reason of debate and an active topic for research. Evolution in adhesive dentistry can be observed from two angles: the first represents the strive to progressively enhance the quality of the products, and the second is the effort made by industry to simplify the process and to reduce the margin of possible human error. The result of the latter is simpler, it has fewer clinical steps, and its manufacturers’ recommendations are less complex and less time-consuming for dental practitioners. One example is the acid etching time of enamel that is 15 s in the instructions of most commercial products [9,10,11,12], which is often lower than the recommended time in the literature that varies between 30 and 90 s. Another example is that, in the instructions of some of those manufacturers, primer application is also recommended on the enamel.

Furthermore, despite the morphological and chemical difference between unground aprismatic enamel and ground enamel, no clear recommendations are given by manufacturers nor by the literature about specific etching procedures for each situation. So far, despite extensive data on the adaptation of adhesive restorations on ground enamel [13,14,15], there is a lack of information about a potential influence of etching time on ground and unground enamel, and the effect of primer application on each of these two surfaces.

In general, three actions have to be reproduced with rigorous attention in order to achieve predictable results [16]: the etching of enamel usually by means of 35–37% orthophosphoric acid, the lowering of surface energy, and the cleaning and infiltration of dentin by means of the priming/bonding system [17,18].

Unfortunately, the substrate on which adhesive procedures have to be performed are not uniform: enamel is mainly composed of inorganic material, 96% hydroxyapatite, while the inorganic fraction of dentin is only 72%. Dentin age and carious or traumatic events influence the diameter of tubules. Enamel is not a homogeneous substrate due to the orientation of hydroxyapatite prisms. A significant difference also exists between morphology of ground and unground enamel. The latter is, in fact, hypermineralized and therefore less prone to the effect of orthophosphoric acid compared to ground enamel [19,20].

The aim of this in vitro study was to investigate the effects of multiple combinations of ground or unground surface, orthophosphoric acid etching time, and primer application on the micro-shear bond strength on enamel. 

The first null hypothesis was that enamel grounding does not affect the micro-shear bond strength. The second null hypothesis was that no bond strength difference exists between enamel acid etching time of 15 and 30 s. The third null hypothesis was that primer application on enamel does not enhance the micro-shear bond strength. 

## 2. Materials and Methods

### 2.1. Teeth Preparation

Thirty-two freshly extracted human maxillary third molars were selected for this study after careful inspection to exclude teeth with caries and visible defects, and they were stored in 0.1% thymol solution. After scaling and cleaning, the roots were removed by cutting horizontally 1 mm under the cemento–enamel junction (CEJ), parallel to the occlusal surface. Each crown was then sectioned in half in the buccolingual direction, and the resulting parts were each fixed on a metallic holder (Baltec, Balzer, Liechtenstein), using light-curing resin, with the section areas in contact with the resin and the approximal surfaces positioned parallel to the plane of the holder. The samples were then randomly assigned to the eight test groups (*n* = 8).

### 2.2. Surface Treatment

The approximal enamel surfaces were submitted to the following treatments, which are also presented in Table 1.

Group unground 15, without primer application (U15): 37.5% orthophosphoric acid (Gel Etchant, Kerr, Orange, CA, USA) was applied for 15 s on intact unground enamel, followed by 15 s water rinsing and subsequent gentle drying with compressed air. A transparent plastic tube of an internal diameter of 1.8 and 3 mm height was then placed over the etched area, and an adhesive layer (Optibond FL Adhesive, Kerr) was applied for 20 s, using a fine brush. A dry brush was then used to eliminate excess resin, and then the adhesive was polymerized by means of a high-power light-curing unit (LCU) (Valo Cordless, Ultradent Products, Salt Lake City, UT, USA). The irradiance of 1000 mW/cm^2^ was measured and controlled at the beginning of each group. A hybrid composite (Harmonize, Kerr) was used to carefully fill the tube. The resin composite was polymerized for 20 s with the same LCU. Group unground 15 with primer application (U15P) followed the same methodology, with the exception of primer application for 15 s, followed by a gentle air blowing after enamel etching and before bonding application. By modifying both etching time to 30 s and rinsing time to 30 s, groups unground, 30 s etch, without primer application (U30), and unground, 30 s etch, with primer application (U30P) were obtained. 

Group ground 15, without primer application (G15): the enamel surface was ground by means of a 40 µm finishing diamond bur (Komet Dental, Lemgo, Germany) under profuse water and an operating microscope. The same procedure as in Group U15 was applied for adhesion and resin composite build-up. Group ground 15, with primer application (G15P) followed the same methodology, with the exception of primer application after enamel etching and before bonding application. By modifying both etching time to 30 s and rinsing time to 30 s, groups ground, 30 s etch, without primer application (G30) and ground, 30 s etch, with primer application (G30P) were obtained.

The plastic tubes were removed immediately after light curing and the samples were examined under an optical microscope (30×) for visible adhesive defects. Only samples without defects were included into the study. They were stored in artificial saliva for 1 week, in an incubator (INP-500, Memmert GmbH, Büchenbach, Germany), at 37°, in order to allow a complete post-polymerization (Figure 1).

### 2.3. Micro-Shear Testing

Samples were mounted in a universal testing machine (Shimadzu AG-X, Shimadzu, Kyoto, Japan), and a semi-circular jig of 1.8 mm diameter was used to apply a shear load on the resin composite cylinders at a crosshead speed of 1 mm/min until failure. The failure load was recorded for each sample, and the fragments were then analyzed through an electronic optical microscope (Keyence VHX-5000, Keyence International, Mechelen, Belgium) to measure the failure area and to determine the failure mode between cohesive in enamel, cohesive in composite or mixed adhesive. The bond strength (MPa) was obtained by dividing the measured failure load by the calculated area for each cylinder. 

### 2.4. Morphological Analysis

Additional samples representing the enamel surface of each group, without applying the adhesive and resin composite, were further analyzed by using a scanning electron microscope (SEM) (Sigma 300VP, Zeiss, Oberkochen, Germany). A qualitative evaluation was made to compare the effect of each etching mode on the surface.

### 2.5. Statistical Analysis

The different average effects of surface type, etching time, and primer application on the bond-strength values were analyzed by using repeated measures ANOVA, followed by Fisher’s LSD post hoc test to assess possible pairwise group mean differences. Shapiro–Wilk’s normality test was used to check for the normality assumptions of ANOVA.

## 3. Results

For samples etched for 15 s, bond-strength values ranged from 8.2 ± 2.1 to 11.3 ± 1.4 MPa, respectively, for U15P and G15. For samples etched for 30 s, bond-strength values ranged from 15.2 ± 2.7 to 19.6 ± 3.0 MPa, respectively, for U30 and G30.

Results in Figure 2 and Table 2 show that there was a statistically significant difference (*p* ≤ 0.05) between enamel surface type (ground vs. unground) and etching time (15 s vs. 30 s). Fisher’s LSD post hoc test results show that G30 is different from U30 and U30P, and that all groups etched for 15 s are statistically different from all the ones that were etched for 30 s, independent of primer application. All failures were classified as mixed adhesive. Representative micrographs of the SEM analysis are presented in Figure 3.

## 4. Discussion

Specifically, in three-step etch-and-rinse adhesive systems, such as Optibond FL, it is recommended to etch enamel and dentin, followed by the application of a primer and an adhesive. In their instructions, most manufacturers do not differentiate between ground enamel present within preparations or beveled margins and unground enamel that is just beyond those margins or on unprepared surfaces, and they recommend a single etching time of 15 s, which is mostly to make the procedure faster and more convenient by aligning with the etching time of dentin. These differences could lead to the assumption that the same etching time of these different tissues should yield different adhesive strengths. Furthermore, the present lack of information about a potential influence of etching time on ground and unground enamel, as well as the effect of primer application on each of these two surfaces, led us to develop this study.

The choice of using third molars was based on the large amount of available extracted teeth, which allowed for a good standardization of the interproximal surfaces’ dimensions, as well as a substantial verticality of the same surface in order to have the most homogeneous possible pool of samples.

The results of this study show a significantly higher bond strength for the extended etching time of 30 s on both ground and unground enamel. Moreover, the SEM morphological analysis (Figure 2) shows that 30 s enamel etching better exposes the hydroxyapatite prisms and yields a better micromorphological pattern, which may potentially be related to an increase in adhesive surface, on both ground and unground enamel. These results are not surprising when the morphology and the chemistry of the superficial layer of enamel is considered. This surface is in continuous contact with external agents such as acidic foods and drinks, which cause demineralization, as well as with re-mineralizing elements, such as calcium and phosphate deriving from common diets (cheese, milk, and yogurts) or fluoride from toothpaste. The result of this alternating process is the presence of a highly mineralized external enamel surface, which is quite resistant to professionally used etching agents, such as orthophosphoric acid, as in this particular case. Another way to increase bond strength is to remove by grinding (bevel) the aprismatic enamel which allows to established adhesion on prismatic enamel.

Clinically, it is challenging to clearly delimit the transition between ground and unground enamel, especially in shallow bevels, and it is therefore preferable to etch enamel slightly beyond the area of presumed transition, to be sure that the uncontrollable excess bond and subsequent composite are well-linked to enamel and to avoid gaps. Insufficient adhesion at this level would make this part of the restoration prone to infiltration of staining agents or even bacteria, possibly leading to marginal discoloration and secondary decays. Furthermore, we would like to underpin that, due to a general tendency to overfill cavities (as it is impossible to fill the cavity exactly to the periphery of the bevel), it is obvious that the margins of most adhesive restorations rest on unground enamel, when enamel is a substrate. This aspect represents the major reason of our study design, as it corresponds to the majority of clinical situations in class I, II, III, IV, and V adhesive composite restorations with enamel margins. 

Another aspect is the application of a primer on enamel. Some manufacturers recommend this step for the following two reasons: the presence of glycerol-phosphate dimethacrylate (GPDM) in the primer, which could contribute to a better adhesion by forming a chemical bond; and for the primer’s capacity of lowering the surface energy, allowing a better bonding wettability. In the present experiment, primer application did not lead to an improved bond strength, but in some cases, it even showed a detrimental effect, as in the difference between groups G15 and G15P. From a chemical point of view, the GPDM adsorbed to hydroxyapatite does not form a stable calcium salt [21]. This weak ionic link can even be easily attacked by hydrolysis in the patient’s mouth, thus possibly worsening the adhesion over time in clinical situations. Finally, the presence of water in a primer could be theoretically detrimental to adhesion, especially in cases where the solvent (ethanol) is not able to completely remove it by evaporation. 

A variety of tests can be used to measure adhesive bond strength, ranging from the macro- to the microscale, and they can either apply tensile or shear loads. An ambiguity still exists around which test is the most relevant to use, and the current recommended one is the micro-tensile [22,23]. However, this test requires delicate sectioning of dental hard tissues, which becomes complex on enamel, since it is a brittle material and thin compared to dentin. A guidance article about bond-strength testing, published in 2017 by the Academy of Dental Materials [23], indicates that micro-shear bond tests are valid for measuring the adhesion on enamel. Therefore, in the present study, the last test was chosen because it is also well established in the literature [24,25], and to limit the number of premature failures. A tube of 1.8 mm diameter was chosen in order to allow for an easier application of the adhesive system compared to narrow tubes under 1 mm, for example. This would also allow us to avoid applying the bonding agent on a larger surface than the composite cylinder, which would also add bias to the results.

In summary, this in vitro study showed higher bond strength and an enhanced micromorphological surface pattern when an extended etching time of 30 s was used on ground and unground enamel. Moreover, ground enamel yielded better results than unground enamel. Primer application on enamel did not show any significant improvement, and it led to a lower bond strength on the ground enamel etched for 15 s.

The limitations of this in vitro study are that the used test was static and that more clinically relevant fatigue and ageing processes or even in vivo studies are needed to confirm the results.

In future studies, it would be worth investigating the effect of diamond bur granulometry on the quality of adhesion to enamel and/or to dentin, both before and after fatigue.

## 5. Conclusions

Within the limitations of this in vitro study, based on micro-shear bond-strength testing, the results showed that surface treatment of enamel by grinding and the duration of etching can influence adhesion. The first and second null hypotheses that neither etching time nor grinding can influence bond strength on enamel were both rejected. Primer application does not seem to have a positive effect on enamel adhesion and, in some cases, can lower the bond-strength values; thus, the third null hypothesis was partially rejected. In future studies, it would be worth investigating the effect of diamond bur granulometry on the quality of adhesion to enamel and/or to dentin, both before and after fatigue.

## Figures and Tables

**Figure 1 polymers-13-03589-f001:**
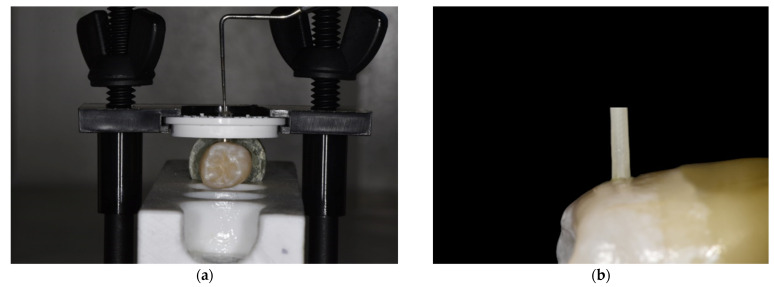
Photographs of the experimental setup (**a**) and of one specimen (**b**).

**Figure 2 polymers-13-03589-f002:**
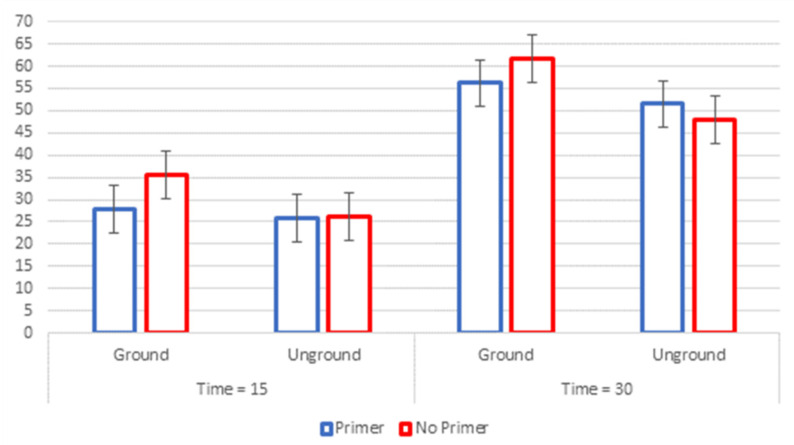
Group means comparison (vertical bars denote 95% confidence intervals).

**Figure 3 polymers-13-03589-f003:**
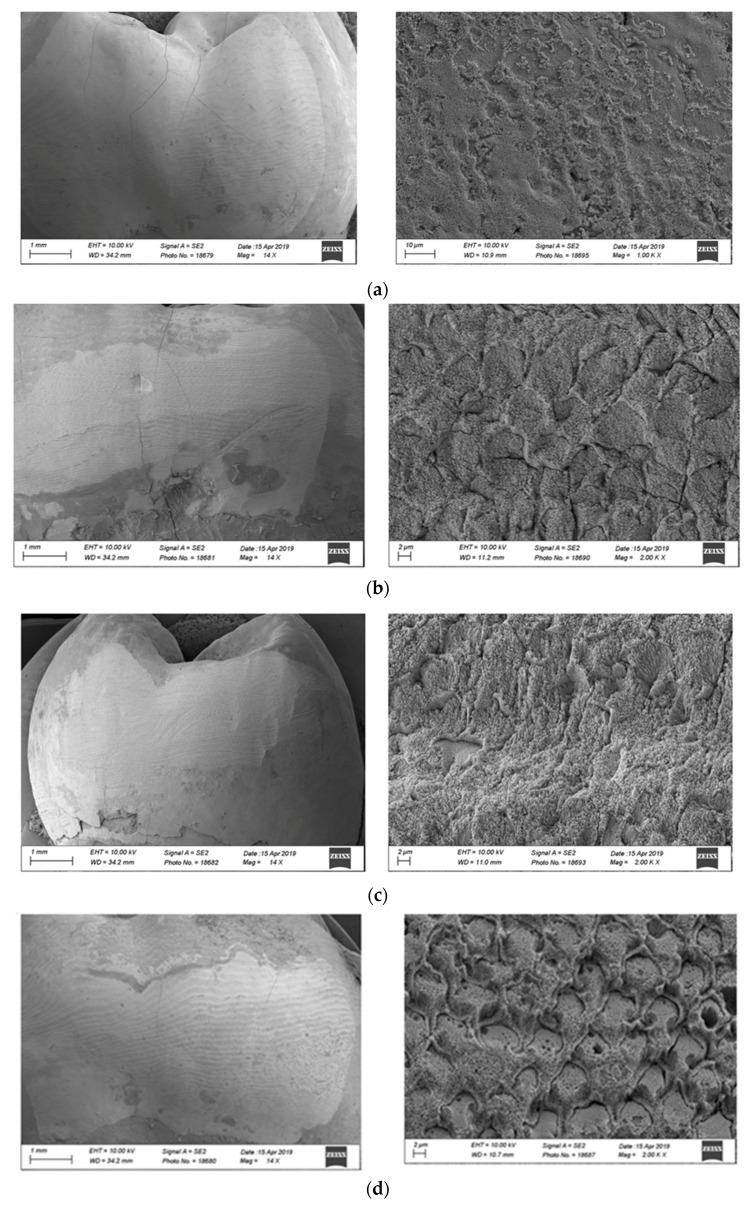
SEM images of different enamel surface treatments. (**a**) (Group U15) Enamel prisms are hardly detectable. (**b**) (Group U30) Enamel prisms are more visible. (**c**) (Group G15) Enamel prisms are more prominent. (**d**) (Group G30) Enamel prisms are very detectable, and bonding area appears to be larger.

**Table 1 polymers-13-03589-t001:** Experimental group details and materials employed.

Group	Surface Type	Etching Time	Bonding and Resin Composite Manufacturer
**U15**	Unground enamel without primer application	H3PO4 15 s (Gel Etchant, Kerr)	Bond (Optibond^TM^ FL Adhesive, Kerr) + Composite resin (Premise Kerr)
**U30**	Unground enamel without primer application	H3PO4 30 s (Gel Etchant, Kerr)	Bond (Optibond^TM^ FL Kerr) + Composite resin (Premise Kerr)
**G15**	Ground enamel without primer application	H3PO4 15 s (Gel Etchant, Kerr)	Bond (Optibond^TM^ FL Kerr) + Composite resin (Premise Kerr)
**G30**	Ground enamel without primer application	H3PO4 30 s (Gel Etchant, Kerr)	Bond (Optibond^TM^ FL Kerr) + Composite resin (Premise Kerr)
**U15P**	Unground enamel Primer application	H3PO4 15 s (Gel Etchant, Kerr)	Bond (OptibondTM FL Adhesive, Kerr) + Composite resin (Premise Kerr)
**U30P**	Unground enamel primer application	H3PO4 30 s (Gel Etchant, Kerr)	Bond (OptibondTM FL Adhesive, Kerr) + Composite resin (Premise Kerr)
**G15P**	Ground enamel Primer application	H3PO4 15 s (Gel Etchant, Kerr)	Bond (OptibondTM FL Adhesive, Kerr) + Composite resin (Premise Kerr)
**G30P**	Ground enamel Primer application	H3PO4 30 s (Gel Etchant, Kerr)	Bond (OptibondTM FL Adhesive, Kerr) + Composite resin (Premise Kerr)

**Table 2 polymers-13-03589-t002:** Micro-shear bond-strength values, standard deviation, and groupings of the different samples.

Treatment Applied	Mean MPA	SD	Grouping				
G30	19.6	3.0	A				
G30P	17.9	1.7	A	B			
U30P	16.4	1.7		B	C		
U30	15.2	2.7			C		
G15	11.3	1.4				D	
G15P	8.8	3.0					E
U15	8.3	1.4					E
U15P	8.2	2.1					E

## Data Availability

Not applicable.

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
