# Peer review of "Effect of Multiple Enamel Surface Treatments on Micro-Shear Bond Strength"

_polymers, 2021, doi:10.3390/polym13203589_

Round 1
Reviewer 1 Report
Dear authors
Even recognizing that a lot of efforts were demanded for producing this manuscript, the issue presented here has already been extensively approached in the dental literature. The results are only confirmatory, thereby not presenting any novelty that encourage its acceptance to be published.
Author Response
Reviewer 1
Even recognizing that a lot of efforts were demanded for producing this manuscript, the issue presented here has already been extensively approached in the dental literature. The results are only confirmatory, thereby not presenting any novelty that encourage its acceptance to be published.
We disagree with this point of view of the reviewer. In reality, no data with our experimental setup is available as the manufacturer recommends a complete different approach, based on 15 s etching of enamel.
Reviewer 2 Report
The authors present an in vitro study evaluating the effect of different surface treatments on adhesion strength. This is an interesting topic, and the manuscript presents the relevant information.
The manuscript misses the lines numbers, which difficulties the review process.
Keywords are missing.
Please describe the thirds molar's provenience. What was the patient´s age? All teeth were erupted?
What was the longest storage time in the thymol solution?
How was the sample size determined? Did the authors perform a sample size calculation?
How was the samples randomization performed?
The primer used was the one from the Optibond FL system?
Regarding the enamel grounding, please provide details for bur contact time and applied force. Was the same burr used for all procedures?
Morphological analysis: how many samples were analyzed per group?
Statistical analysis: what p-value was considered for statistical significance?
I found figure 1 unnecessary since it presents the same information as table 2.
Did the authors acquire some images after the primer application?
I suggest the following information be moved to the introduction section: “Rigorous attention to each step of the adhesive process is mandatory in order to achieve predictable results [16]. In general, three actions are needed for the establishment of a proper dental adhesion: the etching of enamel usually by means of 35-37% orthophosphoric acid, the lowering of surface energy, and the cleaning and infiltration of dentin by means of the priming/bonding system [17,18].”
I suggest the following information be moved to the introduction section: “Enamel is mainly composed of inorganic material, 96% hydroxyapatite, while the inorganic fraction of dentin is only 72%. Dentin age and carious or traumatic events influence the diameter of tubules. Enamel is not a homogeneous substrate due to the orientation of hydroxyapatite prisms. A significant difference also exists between morphology of ground and unground enamel. The latter is, in fact, hypermineralized and therefore less prone to the effect of orthophosphoric acid com-pared to ground enamel [19,20].”
Discussion section: do authors think other adhesives, for instance, self-etch ones, will present similar results?
Discussion section: although implied, the authors fail to clearly state why adhesion on ground enamel is improved. Please clarify it.
Discussion section: the increase in etching time is doable in a clinic situation, but what authors think about grounding sound enamel to increase the adhesion?
I suggest the following information be moved to the discussion section: “In future studies, it would be also worth investigating the effect of diamond bur granulometry on the quality of adhesion to enamel and/or to dentin, both before and after fatigue.”
Author Response
Reviewer 2
Comments and Suggestions for Authors
The authors present an in vitro study evaluating the effect of different surface treatments on adhesion strength. This is an interesting topic, and the manuscript presents the relevant information.
The manuscript misses the lines numbers, which difficulties the review process.
Done
Keywords are missing.
Done
Please describe the thirds molar's provenience. What was the patient´s age? All teeth were erupted?
Thirds molar were fully erupted and caries-free and came from patients ranging of different ages.
What was the longest storage time in the thymol solution?
Up to 1 year
How was the sample size determined? Did the authors perform a sample size calculation?
Yes the size was in accordance with similar studies. A subsequent power analysis was anyway and gave a result above 80%
How was the samples randomization performed?
By means of a randomization trial list.
The primer used was the one from the Optibond FL system?
Yes.
Regarding the enamel grounding, please provide details for bur contact time and applied force. Was the same burr used for all procedures?
No standardization of the contact time was performed, but the contact time was around xxx seconds. The applied force corresponded to regular force applied on a patient.
After the preparation of each tooth, the burs were discarded.
Morphological analysis: how many samples were analyzed per group?
3 samples per group and all 3 gave the same results.
Statistical analysis: what p-value was considered for statistical significance?
The Statistical significance was set at 0.05 (p ≤ 0.05)
I found figure 1 unnecessary since it presents the same information as table 2.
We would like to keep the Figure 1 and Table 2.
Did the authors acquire some images after the primer application?
No
I suggest the following information be moved to the introduction section: “Rigorous attention to each step of the adhesive process is mandatory in order to achieve predictable results [16]. In general, three actions are needed for the establishment of a proper dental adhesion: the etching of enamel usually by means of 35-37% orthophosphoric acid, the lowering of surface energy, and the cleaning and infiltration of dentin by means of the priming/bonding system [17,18].”
Done
I suggest the following information be moved to the introduction section: “Enamel is mainly composed of inorganic material, 96% hydroxyapatite, while the inorganic fraction of dentin is only 72%. Dentin age and carious or traumatic events influence the diameter of tubules. Enamel is not a homogeneous substrate due to the orientation of hydroxyapatite prisms. A significant difference also exists between morphology of ground and unground enamel. The latter is, in fact, hypermineralized and therefore less prone to the effect of orthophosphoric acid com-pared to ground enamel [19,20].”
Done
Discussion section: do authors think other adhesives, for instance, self-etch ones, will present similar results?
We do not want to speculate on a different type of adhesives.
Discussion section: although implied, the authors fail to clearly state why adhesion on ground enamel is improved. Please clarify it.
The aprismatic superficial enamel layer is removed by grinding which allows to established adhesion on prismatic enamel. This concept has benn added into the discussion section
Discussion section: the increase in etching time is doable in a clinic situation, but what authors think about grounding sound enamel to increase the adhesion?
See above.
I suggest the following information be moved to the discussion section: “In future studies, it would be also worth investigating the effect of diamond bur granulometry on the quality of adhesion to enamel and/or to dentin, both before and after fatigue.”
Done
Reviewer 3 Report
Dear authors,
I enjoyed reading your paper. It is well-written, coherent and easy to follow. You chose an interesting topic: this laboratory experiment reflects a contemporary clinical issue and has produced interesting information. Here are some minor suggestions, aiming to further improve the article:
- Add some photographs of your specimens / experimental setup. This would help the reader to better visualise the experiment.
- Include the ANOVA tables in the Results section. The presentation of the results would be more complete.
- The graphs in figure 1 would better be bar charts, not line graphs. Bar graphs are preferable to present the difference between the different settings.
- It would be helpful to provide some more detail/information about:
- Did you noticed any voids in the interface between enamel and composite? How did you check for air bubbles entrapped in the interface or in the bulk of the composite?
- How did you control the depth of the grounding you performed on the enamel? Did it change the curvature of the enamel surface?
Author Response
I enjoyed reading your paper. It is well-written, coherent and easy to follow. You chose an interesting topic: this laboratory experiment reflects a contemporary clinical issue and has produced interesting information. Here are some minor suggestions, aiming to further improve the article:
Add some photographs of your specimens / experimental setup. This would help the reader to better visualise the experiment.
Done
Include the ANOVA tables in the Results section. The presentation of the results would be more complete.
Done
The graphs in figure 1 would better be bar charts, not line graphs. Bar graphs are preferable to present the difference between the different settings.
Done
It would be helpful to provide some more detail/information about:
Did you noticed any voids in the interface between enamel and composite? How did you check for air bubbles entrapped in the interface or in the bulk of the composite?
We meticulously controlled bubbles during the application procedure.
How did you control the depth of the grounding you performed on the enamel? Did it change the curvature of the enamel surface?
No, it did not, we only did a slight bevel as we perform on patients for the adhesive restorations
Round 2
Reviewer 1 Report
Dear authors
Although I am aware that my decision will not please you, I have to maintain my point of view regarding the suitability of the present manuscript for being published in Polymer because I am convinced that the issue presented here has already been extensively approached in the dental literature and that the results are only confirmatory. To sustain my decision, I am quoting three studies in the field that have been published more than 10 years ago.
Pivetta MR, Moura SK, Barroso LP, Lascala AC, Reis A, Loguercio AD, Grande RH Bond strength and etching pattern of adhesive systems to enamel: effects of conditioning time and enamel preparation. J Esthet Restor Dent. 2008;20(5):322-35
Perdigão J, Gomes G, Lopes MM. Influence of conditioning time on enamel adhesion. Quintessence Int. 2006 Jan;37(1):35-41.
Barkmeier WW, Erickson RL, Kimmes NS, Latta MA, Wilwerding TM. Effect of enamel etching time on roughness and bond strength. Oper Dent. 2009 Mar-Apr;34(2):217-22
Author Response
Dear reviewer, we appreciate the time you have dedicated to our paper again, but we still believe that the specific parameters of our study are not “extensively approached” in the literature as claimed. We understand that notions might appear covered by the literature, we also had this perception before launching the study, but we were not able to find one paper that groups the same factors that we are directly comparing, also by following the latest recommendations for bond strength testing. So we decided to address it in this paper.
Perdigão J, Gomes G, Lopes MM. Influence of conditioning time on enamel adhesion. Quintessence Int. 2006 Jan;37(1):35-41.
In this paper they do not use OptiBond FL which is the golden standard, and which was the focus of our paper, because it is still is the most used 3-step adhesive system by clinicians and even in the literature. They did not even compare ground to unground, nor primed to non-primed (which in our opinion is the main particularity of our study as priming enamel is still debated) which means that this study is completely different from ours. In that same paper, the authors used the microtensile test which in itself changes a lot and could introduce defects in the interface of brittle materials like enamel as specified in the Academy of Dental Materials Guidance on microtensile bond testing (article from 2017, reference 23 in our paper):
“Micro-tensile bond strength testing on enamel is more difficult than on dentin due to the relatively brittle nature of enamel. A macro- or micro-shear [27] bond strength test is an alternative approach that requires less specimen processing and thus reduces the possibility of introducing surface or edge defects”
Wayne W Barkmeier 1 , Robert L Erickson, Nicole S Kimmes, Mark A Latta, Terry M Wilwerding. Effect of enamel etching time on roughness and bond strength. Oper Dent Mar-Apr 2009;34(2):217-22. doi: 10.2341/08-72.
In this paper similar comments to the previous study as it is also significantly different from ours, especially with the no use of OptiBond FL, no use of three-step adhesive systems even, no comparison between ground/unground, primed/unprimed. Additionally, the etching time they investigated for the bond test was either 15 s and 60 s. The 30 s they present in the table was only performed for the roughness measurement. It is therefore a completely different study.
MARCELA RIGUERA PIVETTA et al. Bond Strength and Etching Pattern of Adhesive Systems to Enamel: Effects of Conditioning Time and Enamel Preparation
This article still remains different from ours because they did not investigate the effect of priming or not priming enamel. Moreover, they used a flowable composite, and we know from the literature that in such tests, there’s a correlation between the E-modulus of the composite that is used and bond values, and this is why in our study we used a clinically comparable condition with regular consistency hybrid composite. The previously mentioned elements added to some other operatory differences (we light-cured the composite for 20 s instead of 40 s also reflecting the clinical situation), and the fact that they did not evaluate the effect of priming enamel, makes us believe that the element of novelty is in fact present in our paper, and that we provide a direct comparison of all these important factors.
Round 3
Reviewer 1 Report
Dear authors,
Good arguments!